# Automated Quantitative Analyses of Fatigue-Induced Surface Damage by Deep Learning

**DOI:** 10.3390/ma13153298

**Published:** 2020-07-24

**Authors:** Akhil Thomas, Ali Riza Durmaz, Thomas Straub, Chris Eberl

**Affiliations:** 1Fraunhofer Institute for Mechanics of Materials, 79108 Freiburg im Breisgau, Germany; thomas.straub@iwm.fraunhofer.de (T.S.); chris.eberl@iwm.fraunhofer.de (C.E.); 2Department of Microsystems Engineering, University of Freiburg, 79110 Freiburg, Germany; 3Institute for Applied Materials Computational Materials Science IAM-CMS, Karlsruhe Institute of Technology, 76131 Karlsruhe, Germany

**Keywords:** deep learning, semantic segmentation, extrusions, micro cracks, slip trace analysis, generalization

## Abstract

The digitization of materials is the prerequisite for accelerating product development. However, technologically, this is only beneficial when reliability is maintained. This requires comprehension of the microstructure-driven fatigue damage mechanisms across scales. A substantial fraction of the lifetime for high performance materials is attributed to surface damage accumulation at the microstructural scale (e.g., extrusions and micro crack formation). Although, its modeling is impeded by a lack of comprehensive understanding of the related mechanisms. This makes statistical validation at the same scale by micromechanical experimentation a fundamental requirement. Hence, a large quantity of processed experimental data, which can only be acquired by automated experiments and data analyses, is crucial. Surface damage evolution is often accessed by imaging and subsequent image post-processing. In this work, we evaluated deep learning (DL) methodologies for semantic segmentation and different image processing approaches for quantitative slip trace characterization. Due to limited annotated data, a U-Net architecture was utilized. Three data sets of damage locations observed in scanning electron microscope (SEM) images of ferritic steel, martensitic steel, and copper specimens were prepared. In order to allow the developed models to cope with material-specific damage morphology and imaging-induced variance, a customized augmentation pipeline for the input images was developed. Material domain generalizability of ferritic steel and conjunct material trained models were tested successfully. Multiple image processing routines to detect slip trace orientation (STO) from the DL segmented extrusion areas were implemented and assessed. In conclusion, generalization to multiple materials has been achieved for the DL methodology, suggesting that extending it well beyond fatigue damage is feasible.

## 1. Introduction

The ongoing digital transformation of materials pursued by the community (e.g., Materials Genome Initiative [1] in the US and related German efforts MaterialDigital [2] and NFDI4MSE [3]) aims to enable accelerated product development and predictive maintenance. These objectives rely on digital representations of materials, also referred to as digital material twins, along its value chain and product lifetime. Further, achieving these goals demands that the digital material representations reproduce the evolution of materials microstructure and properties in its underlying data- or knowledge-driven models.

Advances in material modeling and computational power allow increasingly realistic predictive simulations. To some extent, the quality improvement can be ascribed to a paradigm shift from conventional elasto-plastic simulation methods to the crystal plasticity finite elements (CPFE) and related fast Fourier transform (CPFFT) models. These methods feature microstructure-induced anisotropy and strain localization. Since fatigue poses a common failure cause, extensive efforts have been made to model fatigue crack initiation and growth utilizing these models, e.g., [4]. These fatigue-related models transcribe microstructural strains and stresses into so-called fatigue indicator parameters (FIP) measuring the local vulnerability to crack initiation under cyclic load. Especially, when accurate prediction of such early fatigue states is concerned, a thorough understanding of the underlying fatigue mechanisms is required in order to account for them in modeling. This applies particularly to the high- and very high-cycle fatigue regime (HCF/VHCF) where the microstructure governs the localization of damage and cracks as well as their evolution. In these regimes the global lifetime of a specimen is to the largest extent determined by initial fatigue states such as accumulation of plasticity, crack initiation as well as micro and short crack growth. However, incomplete knowledge of early fatigue mechanisms and pronounced computational cost of CPFE methods necessitate a variety of assumptions and simplifications. Thus, there is a need for statistically validating the predicted microstructural degradation response of material models in the initial fatigue stages.

Validation of micromechanical fatigue models requires information on localization of damage within the microstructure as well as temporal information on its emergence. Furthermore, experimental information on predominant slip trace orientations is necessary. The latter requirement is justified by the impact of relative orientation between slip trace and grain boundary trace, where orthogonality between these traces promotes crack initiation [5]. Moreover, identifying slip system candidates is of relevance for validation, since some modeling approaches entail slip system dependent hardening laws [6].

In literature, prevalently high-resolution digital image correlation (HR-DIC) was employed to validate CPFE strains in SEM images of small grain ensemble, e.g., [7,8,9]. Most HR-DIC related validation approaches focus on slip band emergence in the low-cycle fatigue regime, rather than crack initiation and micro/short crack growth in the HCF/VHCF regime. However, the emergence of slip bands does not necessarily lead to the formation of critical extrusions and cracks. In order to validate the capability of a fatigue micromechanical simulation model to predict events such as crack initiation, a complementary validation concept is required. This concept should rely on robust localization of extrusions and cracks. Another demand that is made to the approach is a high degree of automation as statistical validations are required and the materials as well as mechanisms are diverse.

The hereby presented work tackles the challenge of pixel-wise damage localization to identify surface residing extrusions and cracks (i.e., semantic segmentation) in secondary electron (SE2) SEM images with automated data-driven methods. Moreover, determining slip trace orientations (STO) from the segmented extrusions is addressed.

Over the past decade, deep learning (DL) approaches to the semantic segmentation problem have gained popularity. DL is a type of representation learning, where the feature selection is carried out automatically from the data. An important factor behind the success of DL techniques is the active development of network architectures that could capture and utilize features from images effectively. In the context of this work, features arise from surface topography. Additionally, deep networks have the capability to become invariant to the changes in images caused by brightness, contrast, sharpness changes, etc., given they are provided with enough data. These characteristics motivated our choice setting our scope on DL over conventional machine learning (ML) or image processing techniques for the task of segmenting our SEM images containing substantial variance. Another reason for its success is the availability of annotated data sets and the associated semantic segmentation contests. PASCAL Visual Object Classes (VOC) [10] Challenge with 21 classes and 2913 annotated images is arguably the most popular among them. Other notable data sets include Microsoft COCO [11], Cityscapes [12] and Synthia [13]. In regard to data-driven methods, a challenge frequently faced in the materials science community is the shortage of labeled images. This applies to cracks and extrusions from SEM images as well, as there were no public labeled data sets for segmentation available. Hence in this work, two distinct steel data sets and one copper data set with pixel-wise labels for the classes crack and extrusion in SE2 SEM images were created, satisfying a prerequisite for learning. The biomedical community also faces similar issues concerning labeled data shortages, specifically in terms of image segmentation. This has led to the development of the U-Net architecture [14], which is able to cope with small data sets.

In the materials domain, classical ML techniques are now prominently being used in materials discovery. Deep learning (DL) techniques are also gradually emerging in the domain. For instance in [15], a steel phase classification in dual-phase steels employing a fully convolutional neural network was successfully implemented. Furthermore, there is work on steel defect detection [16]. However, to our knowledge, the DL segmentation of micro cracks, short cracks, and extrusions as well as automatic slip trace orientation determination from extrusions have not yet been attempted.

The reliable and automated damage segmentation was identified as a crucial step to statistically validate and improve micromechanical material models. As a result, digital material representations can be generated, enabling accelerated product development. The vast material and microstructure landscape with its various fatigue mechanism has made automation essential. Apart from automation, this obviously places the requirement of transferability across material domains on the segmentation model. Hence, emphasis is placed on generalizability of the segmentation model to other materials as well as applicability to images captured by other SEM operators. In the following, we show how a data-driven model that can generalize to this variance can be derived from the surface damage information contained in fatigued specimens.

## 2. Method

The method section is preluded with a description of the created data sets incorporating extrusions and cracks from three materials. ranging from microstructurally short cracks to long cracks. Subsequently, the methodology for the segmentation of cracks and extrusions is presented. Data augmentation techniques adopted to accommodate the lack of labeled images and to simulate variance in the data set are introduced. The U-Net architecture, loss functions and hyperparameter optimizations while training are introduced. Concluding the method section, conventional image processing methods to determine STO from the segmented extrusions are presented.

### 2.1. Data Set

#### 2.1.1. Data Set Creation

This section does not cover the fatigue process nor the loading conditions, since this is described elsewhere [17], but rather focuses on the establishment of the data sets for learning the pixel-wise damage localization. The training and testing data sets were generated from stitched SE2 images of micro samples in the highly loaded regime after fatigue (Figure 1) and its annotated correspondences.

Acquisition of the stitched images was performed by two operators on an automatized Zeiss Supra 40VP SEM (Carl Zeiss Microscopy GmbH, Oberkochen, Germany), while the annotation process included a single expert drawing the perceived border of damage locations and assigning a class. During image acquisition with a surface-sensitive Everhart-Thornley SE2 detector, an acceleration voltage of 20 kV, working distances varying from 10 to 14 mm, magnifications ranging from 1500× to 2200× and an image resolution of 2048 × 1536 were applied. This was required to capture the relevant features of the extrusion and micro crack classes, enabling the network to differentiate between background and foreground classes. Consequently, large stitched images typically in the order of 15,000 × 30,000 pixels were obtained. Thus, to avoid graphic processing unit (GPU) memory limitations the image is, as proposed in [14], passed to the network in batches of 760 × 760 pixel tiles. By virtue of the applied U-Net architecture (see Section 2.2.2) the mask tile size was reduced in comparison with the input tile. Therefore, prior to extracting the tiles, as apparent in Figure 1a, mirror padding was applied at the sample image boundary. Furthermore, the tiles were extracted with an appropriate overlap to account for the reduced mask tile size and to utilize the data entirely.

The aforementioned routine was applied to create data sets for three materials covering a range of distinct microstructures and fatigue mechanisms. These materials exhibiting different equiaxed and hierarchical microstructures as well as face- and body-centered cubic elementary cells represent a subset of the microstructural diversity. This supports the objective to investigate the feasibility of damage detection generalizing to a wide range of common engineering alloys and crystal structures. Each of these material domains contains extrusions and potentially cracks with unique image textures. As a result, the performance of a trained network to segment damage in an alternate unseen material (domain generalization) and conjunct training with multiple materials could be evaluated.

#### 2.1.2. Data Set Overview

A comparison of data sets is illustrated by a few characteristic data set metrics describing the distribution of extrusion and crack classes in each material data set. The metrics are the number of stitched images N, number of tile images N${}_{t}$, image/pixel percentage for both damage classes p${}_{x}^{y}$ and 80% percentiles of the pixel area ratio for both damage classes P${}_{x}^{p80}$. Subscripts *x* denote the damage classes crack *c* and extrusion *e*, while the superscripts *y* denote whether it is an image *i* or pixel *p* percentage value. For instance, the value P${}_{e}^{p80}$ = 10% for the ferrite testing data indicates that 80% of the images have an extrusion area percentage of 10% or less.

From Table 1 and the p${}_{c/e}^{p}$ values, the substantial intrinsic class imbalance of the three data sets becomes obvious. Fatigue loading in the HCF/VHCF regime leads to degradation of only a few critical grain clusters. Thus, only a small portion of the pixels contains damage. Moreover, Table 1 illustrates that there are more labeled ferritic samples leading to an additional imbalance between the data sets. When training on combined data sets is concerned, this needs to be accounted for.

#### 2.1.3. Data Set i: Extrusions and Cracks in Ferritic Steel

Initially, the damage observed in a ferritic steel EN 1.4003 with a body centered cubic (bcc) crystal structure was investigated. Training tiles originated from multiple samples, which were, as opposed to the other materials, fatigued with differing strain amplitudes. Cracks ranging from microstructurally short cracks to long ones were observed in the material. As apparent in Figure 2a–f typical cracks (green) present in the data set show distinct contrast compared to the background and frequently are encompassed by extruded volume (blue). This extruded volume is either present due to the accumulation of plasticity prior to crack initiation (Figure 2f) or resulting from stress concentration in the vicinity of the crack tip (e.g., Figure 2c). While in micromechanics the term extrusion refers to the accumulation of dislocation steps and vacancies at a surface forming a unique topography before crack nucleation, here both cases are merged into the extrusion class. Even though extrusions, according to [18], can be classified based on their shape in ribbon-like, tongue-like and macroscopic protrusions superimposed with former types, the ferrite data set contains almost exclusively protrusions. This can be ascribed to the low defect density in the ferritic material and concomitant unimpeded movement of dislocations. Some extrusions are fissured and even contain a multitude of tiny cracks. The data were not annotated to distinguish between these extrusion classes. It is noteworthy that the labels do not exhibit pixel level accuracy since the boundaries of extruded areas in ferrite are faded and thus not always evident.

Apart from cracks and extrusions, shallow scratches and residual particle agglomerates from colloidal silica polishing clients (OP-S) exist at the sample surface. Furthermore, in the course of crack growth, debris caused by friction of the crack surfaces is driven out of the crack obscuring relevant areas. Since the ferrite tiles originate from a comparatively large number of samples, they contain a substantial brightness variance owed to the brightness and contrast settings during acquisition, see Figure 2. Moreover, the fact that the surface of the micro samples is slightly bent leads to a brightness gradient resulting in an additional brightness variance among tiles. This is attributed to the surface orientation dependence of the SE2 detector and leads to concealed extrusions or cracks in over- or under-exposed regions at the sample edge. The sample curvature induced brightness changes occur in all three data sets to some extent. Another feature, apparent for instance in Figure 2f, is the sudden gray-value jumps that can be ascribed to grain boundaries. These pose a frequent location for damage to emerge [19].

#### 2.1.4. Data Set ii: Extrusions in Copper

Additionally, a data set with polycrystalline oxygen-free high conductivity (OHFC) copper exhibiting a face centered cubic (fcc) crystal structure was created. Due to its low yield strength, absence of defects, and the applied loading conditions, extrusions observed in this data set are extensive and spread across grain boundaries (not depicted). The extrusion class entails instances where slip bands extend nearly across the whole sample width (for a tile of which, see Figure 2g) as well as moderately localized protrusions comparable in scale to the ones observed in ferritic steel (Figure 2h). Moreover, multiple fine tongue-like extrusions can be observed, see the right border of Figure 2g. Former larger slip bands observed in copper do not show unique well-defined slip trace orientations but rather indications of wavy slip and multi-slip.

As opposed to the other data sets, the copper data set in the context of this work was utilized solely for testing purposes.

Furthermore, in contrast to the remaining data sets, the surface finish of the copper material was carried out with an electropolishing step. Therefore, the characteristic grain structure induced image background texture is absent and waviness in terms of background topography is noticeable in some sections of the surface, see Figure 2h.

#### 2.1.5. Data Set iii: Extrusions in Martensitic Steel

Concluding the data sets, a complementary extrusion data set comprising extrusions in a steel EN 1.7228 with a martensitic microstructure was prepared. The lattice distortions and the hierarchical microstructure of the material result in confined extrusions, as apparent in Figure 2i,j. In contrast to extrusions observed in the ferritic material, these extrusions can be classified as tongue-like and are one order of magnitude smaller. Similar to ferrite, the sample surface contains OP-S particle agglomerates. Though in contrast to ferrite, extrusions and these polishing artifacts are at a similar scale. Another feature of the images owed to the hierarchical microstructure is the distinct background texture. Since the martensitic microstructure exhibits much smaller effective grain sizes than ferrite, the microstructure variance in a given area is comparatively more pronounced. This leads to the fact that more martensite images contain extrusions despite the lower number of total extrusion pixels, see Table 1.

### 2.2. Semantic Segmentation

#### 2.2.1. Data Variance and Augmentation

The appearance of extrusions and cracks occurring in surface-sensitive SE2 SEM images can differ due to several factors including material, sample fabrication, loading condition during fatigue, SEM operator subjectivity and SEM device specifics.

As observed in the previous section, the microstructure of materials can directly affect the shape and size of extrusions and cracks. The versatility of microstructures employed in engineering leads to a substantial variance in extrusion and crack geometry that cannot be covered by the three treated materials. For instance, plasticity in Nickel samples has shown to culminate in several parallel but distinctly separated extrusions delimited by grain boundaries [20], which was not observed in any of the materials covered in this work. The fabrication of the specimen or components affects the background texture as well as the emergence of fatigue defects. Loading conditions e.g., axiality mainly determines the slip characteristics [21] and thus the shape of the extrusions. Furthermore, slip markings have shown to form a cyclic load induced oxide layer at their surface, which significantly exceeds native oxides in thickness [22]. This can affect the representation of extrusions in SE2 images. During image acquisition, SEM operators are responsible for effects such as defocus, astigmatism, altering brightness, contrast, and ensuring electrical conductivity—influencing the signal to noise ratio. However, in contrast to the vast majority of data sets, e.g., for autonomous driving, the imaging conditions in microscopic images are stable in regard to a consistent viewpoint, minimal occlusion and irrelevance of environmental influence factors.

Our approach to cope with and generalize to the substantial variance was to train the network with an adapted data set. The aim is to adapt the data set in a way that it is a more comprehensive representation of the data that is to be evaluated, even potentially extending to related material domains. To mimic the variance described above, the data sets were artificially augmented by applying various intensity-based and spatial transformations which can be summarized as warping transformations [23]. This process is typically referred to as data augmentation. Additionally, data augmentation increases the tendency to learn without overfitting, alongside network-focused regularization methods such as dropout and batch normalization. The employed augmentation types are described in Table 2. These data augmentations were added to an image augmentation pipeline and applied to the input images with the specified probability on the fly prior to training.

The objective behind introducing motion blurring was to imitate the directed distortion arising from astigmatism. Elastic transformation was incorporated to replicate the variance introduced by different materials and distortions present in the data. Each augmentation threshold was chosen such that the characteristics of the defects were still recognizable and the labels were preserved.

#### 2.2.2. Architecture

The architecture used is a four-level U-Net multiclass segmentation model implemented in PyTorch and depicted in Figure 3. This architecture was originally developed by [14] and utilized for biomedical tasks such as cell segmentation or segmentation of computer tomographic sections. Its fairly shallow structure enables the training with comparatively fewer training examples.

The architecture comprises of a contracting path (encoder) followed by an expanding path (decoder). In the encoder part, each level consists of layers of cascaded convolution operations that extract features (edges, textures, context, etc.) and max pooling that downsamples them. Therefore, the encoder part can be thought of as a hierarchical way of finding features. However, at the end of the encoder layers, the spatial information about the feature is poor, meaning we cannot know where in the image a particular feature is extracted from. This spatial information is then regained using the decoder section. The decoder section of U-Net is similar to the encoder part, with max pooling layers replaced by up-convolution layers. Additionally, at each decoder level, the up-sampled features are concatenated with the corresponding features from the encoder, which enables it to retrieve lost spatial information thus leading to precise localization.

The final layer of the network has three channels, corresponding to background, crack, and extrusion. Different input tile sizes can be used in this architecture, after making sure that each down-sampling is performed on features with even x-y shapes.

### 2.3. Slip Trace Detection

This section presents the methods devised for the detection of two-dimensional slip trace orientation (STO) from the segmented extrusions, namely:**CSO**: Clustering of Scharr gradient based edge orientations.**CHL**: Clustering Hough lines from edges.**ICEO**: Iterative clustering of edge orientations.

Since multiple slip can occur or extrusions can propagate through grain boundaries, the methods need to account for multiple STOs in an extrusion area. This is addressed based on the assumption that when there are multiple STOs in an extrusion, we will find spatially separated groups of parallel slip traces (e.g., extruded regions). On the other hand, when parallel slip traces are not spatially clustered, they can be considered as part of an extrusion with a single STO, given their orientations are within a permitted range. The methods designed consist of two main steps—extraction of prominent slip traces from the segmented extrusion area, and clustering these traces based on orientation and positional features.

In the initial steps, all methods are supplied with the SEM image, and the corresponding extrusion mask (along with its probabilities) predicted by the U-Net. This includes cropping images corresponding to extrusions from the bounding box of the extrusion mask as shown in Figure 4a,b. In Figure 4a multiple extrusions detected from the image can be observed. The analysis was performed for each of them, given they are above a size threshold. Moreover, the extrusion masks enable us to consider only the extrusion area in the STO analysis, rather than the whole bounding box. This is required in instances where there are other high gradient magnitude regions nearby the extrusions, see Figure 4b. Additionally, a strict threshold on the extrusion probability mask was employed by the techniques to further exclude regions of falsely predicted extrusion regions, see the yellow region in Figure 4c.

The CSO method detects the prominent slip traces based on the gradients computed using Scharr kernels. Additionally, it relies heavily on the predicted extrusion probabilities from the U-Net, assuming that extrusion predictions of higher confidence from the network would localize to the desired regions. A higher threshold value was used on the extrusion probability mask. This high confidence region was then convolved with a Scharr kernel to compute the gradients along both image axes, and subsequently, the pixel-wise gradient magnitude and orientation, as depicted in Figure 5a,b, respectively. In the following, the gradient magnitude image was thresholded yielding a binary mask of a further reduced region containing only pronounced gradients. This binary mask extracts gradient orientation blobs at corresponding high gradient magnitude regions. These blobs were then filtered based on their eccentricity, size, and standard deviation of gradient orientation, selected blobs represented in red boxes in Figure 5c.

Clustering was then performed on these filtered gradient orientation blobs. The blobs were represented in a 3-D feature space comprising of median orientation and the x-y coordinates of its centroid. These 3-D points were then clustered using a hierarchical clustering algorithm. The clustering uses a combination of orientation and centroids for distance computation while giving more weight to orientation. The major orientations were then ranked on the basis of the number of points in the clusters.

The Hough method finds prominent slip traces using Canny edge detection [24] followed by Hough transform (HT). The threshold on extrusion probability mask was not used as strictly as in CSO. Motivated by the fact that slip traces appear brighter than its surroundings in SE2 SEM images, morphological dilation was performed prior to Canny to remove edges of darker regions in the image. Subsequent Canny edge detection poses a multi-step algorithm that uses a Gaussian filter for smoothing the image, followed by gradient-based thresholding and refinement. The method also uses further morphological operations—skeletonizing followed by the removal of skeletons with lower pixel count, to refine the Canny edges. An extrusion overlaid with the refined skeletonized edges is illustrated in Figure 5d. Subsequently, HT was performed on the binary skeleton image yielding the corresponding distribution in Hough-space. HT is typically used to detect straight lines and collinear line segments in images, and is employed in various applications including Kikuchi line detection in electron backscatter diffraction (EBSD) patterns [25]. The transformation uses Hesse normal form to represent a line in polar coordinates. It computes all possible lines for each pixel in the edge image, which will be a sinusoidal curve in the Hough-space. Each point in the Hough-space represent a line, and its gray-scale value is computed using an accumulator that counts the number of points in the skeleton that lie on or close to this line. The method then searches through the Hough space and selects top n lines based on the Hough accumulator value, as shown in Figure 5e. The motivation being that most of the prominent lines we get will correspond to the prominent collinear regions in the edge image, which in turn will be along the most prominent orientations in the extrusion.

Parameters of Canny such as Gaussian filter size, and gradient magnitude upper and lower thresholds were optimized for the data set. The underlying motivation is that the edges detected correspond to boundaries of the prevalent protruded segments in the extruded region.

The Hough lines are then clustered together based on the angle of the selected lines, see Figure 5f. A disadvantage of representing slip traces as lines is that their positional closeness can not be inferred. This holds true since there is no link between Hough lines and particular edge segments. Hence, spatial features were not included in clustering. The clusters are then ranked based on the sum of Hough accumulator values for the lines within a cluster. Weighted mean of the orientations was found out for the top three prominent clusters.

The ICEO method selects prominent slip traces from the segmented extrusion area using only Canny and morphological operations. All steps until getting a refined skeleton image are similar to the Hough method. Disjoint regions are then selected from this skeletonized image and filtered based on its eccentricity, allowing only regions approximating lines. These filtered regions have orientation, length, and cluster centroids as features. They are then clustered initially on the basis of only orientation using a strict threshold, as shown in Figure 5g. This initial clustering detects groups of slip traces that are nearly parallel. Then, a silhouette coefficient was computed for each pair of these parallel regions using only their centroids as features, resulting in a silhouette matrix. The silhouette coefficient is a measure of the clustering performance, and the matrix obtained in the step above gives a measure of how well the detected parallel regions are clustered spatially. Then iteratively, the least spatially clustered regions are merged to a single cluster until all the cluster combinations have silhouette coefficient above a threshold value. The clusters are then ranked based on the accumulated length of the regions in the cluster. The clusters after iterative merging are shown in Figure 5h.

## 3. Results and Discussion

Three distinct materials were selected to represent a wide range of engineering alloys. These were utilized to investigate the generalizability of the deep learning methodology delineated in Section 2.2. The materials introduce variations in their damage surface topography. These variations arise due to different dislocation slip characteristics ingrained into the materials through the alloying composition and heat treatment resulting in specific crystal- and microstructures. Aside from material intrinsic sources of variance, the operating and imaging conditions affect the appearance of damage location micrographs. To investigate the semantic segmentation performance different training sets were used and the results were compared.

Having micrographs along with predicted annotations allows us to identify possible candidates of the underlying slip systems. Predominant in-plane orientations of extrusions and intrusions can be extracted by different image processing based methods, shown in Section 2.3, which will be compared.

### 3.1. Semantic Segmentation

This section presents the results of two sets of experiments covering the semantic segmentation of fatigue-induced extrusions and cracks. Initially, U-Net models trained only with augmented ferrite data sets were evaluated on ferrite (source) as well as on martensite and copper (targets). Thereby the source domain performance and the material domain generalization, determining the applicability to a multitude of materials, were evaluated, respectively. In the course of this, the influence of the augmentation configuration on the performance was investigated. Subsequently, with the optimized augmentation pipeline, models were trained in a multi-domain setting on both steel data sets and tested on all data sets. Here, various imbalance correction strategies were applied and the performances of the resulting model evaluated.

#### 3.1.1. Source Domain Performance and Domain Generalization

We had hypothesized in Section 2.2.1 that data augmentation can help mimic various expected variances in data. It is interesting to verify this hypothesis by evaluating at first how different data augmentations affect source domain performance. Additionally, the hypothesis can be substantiated by evaluating how well a model can generalize to multiple material domains. There is a pool of approaches tailored to improve generalization across domains. These include multi-domain training, transfer learning and domain adaptation. Which approach amongst these to use depends on the availability of data as well as the (material) domain gap. Despite potential improvement of performance in the target domain associated with these approaches, most of them require elaborate labeling or retraining efforts. Moreover, extrusions and cracks in different materials share common high-level characteristics. Thus, for the prevalent problem of damage detection, the question arises, if application of a source domain trained network on images of unseen target domain can yield satisfactory results. Is the domain gap small enough for training-time data augmentation alone to provide generalization across material domains and other previously discussed variance in the data?

These questions were addressed by training a U-Net using multiple combinations of augmentations on the ferrite domain, and testing the models on all three domains. The impact of introducing individual augmentation types from Table 2 to the augmentation pipeline was tested and the results are illustrated in Table 3. During training, the loss functions weighted cross entropy loss, dice loss and focal loss were tried out. Since the latter resulted in the best segmentation results, experiments presented were trained with this loss function. Batch normalization or other types of normalization techniques were not applied. Results are evaluated utilizing the metric mean intersection over union of extrusions (mIoU${}_{e}$) and cracks (mIoU${}_{c}$) in ferrite and of extrusions in martensite and copper. Note that the background mean intersection over union is neglected as it is approximately unity in every case. Nevertheless, the overall mean intersection over union (mIoU${}_{o}$) for ferrite is given. In order to keep the training time maintainable the images with a total pixel count of N in the training set were resized by a factor of four.

Here it becomes apparent that the augmentations delineated in Table 2 impact the performance of the network in distinct ways for the three test data sets. Note that relative comparisons between single augmentation types are difficult to conduct since the augmentation probabilities according to Table 2 differ.

**Source domain performance:** In the ferritic test data set, it can be observed that brightness (#1) and Gaussian noise (#5) augmentations cause detrimental effects on the overall mIoU when compared to the experiment with minimal augmentations (#8). Taking into account only the models trained with down-sampled data sets, an increase of ΔmIoU${}_{c}$ of 0.06 and ΔmIoU${}_{e}$ of 0.14 for the evaluation on ferrite was achieved employing a custom set of augmentations (comparing #10 with #8). This custom set included only augmentations that optimized performance on the ferrite and martensite domains. In Section 3.1.3 this optimized model is evaluated.

The brightness augmentation being adverse, could be ascribed to the fact, that extrusions/intrusions or cracks often exhibit extreme intensity values. Their sharp edges cause substantially higher or lower secondary electron emission rates. Additionally increasing or decreasing the brightness through augmentation results in an information loss in the high or low intensity regime concealing the features of the damage locations (further). This applies in particular to a notable amount of damage spots which are located at sample boundaries. These typically suffer from over- or under-exposure owed to the sample surface orientation relative to the SE2 detector. The Gaussian noise especially reducing the ferrite mIoU${}_{c}$ can potentially be attributed to features of narrow cracks not being preserved when the noise is applied. This can be worsened by downsampling.

**Domain generalization to copper:** On a first glance it can be inferred that the models trained on ferrite can be generalized seamlessly to the copper domain as mIoU${}_{e}$ values for copper surpassed the source domain mIoU${}_{e}$ for some models. However, upon detailed analysis, it becomes apparent that the tongue-like extrusions, which constitute a small damage area proportion in copper, are segmented erroneously. This manifests in poor prediction in terms of connectivity as multiple tongue-like extrusions in close vicinity are predicted as a single connected one. Moreover, it is noteworthy that every model involving contrast showed a comparatively poor performance for copper among the individual augmentations.

The superior performance observed in copper for some models can be explained by copper extrusions exhibiting more distinct boundaries to undamaged areas. Meanwhile, the pronounced performance drop in contrast augmented models implies that these models become invariant to contrast changes thus failing to make use of the distinct boundaries in copper extrusions. This feature is particularly important due to the absence of directional features in the copper extrusions i.e., no clear slip trace orientations are present as opposed to ferrite extrusions. Further, there are some limiting factors for domain generalization of ferrite trained models to copper. Since there were virtually no tongue-like extrusions present in the ferrite data set unlike in the copper data set, their accuracy was moderate. The merging of close extrusions and in particular tongue-like extrusions in copper can be ascribed to two characteristics of the damage in ferritic steel. Namely, on the one hand the absence of distinct extrusion boundaries and on the other hand the predominance of extensive protrusions.

**Domain generalization to martensite:** Meanwhile, it can be seen that domain generalization to martensite was unsatisfactory. Despite certain augmentations leading to an increase of 0.08 on martensite, the final absolute mIoU${}_{e}$ of 0.14 is quite poor. Also, in contrast to the observations in ferrite, the martensite mIoU${}_{e}$ increases when Gaussian noise is applied.

Gaussian noise improves the performance of the ferrite trained model on the martensite data as latter exhibits an inferior signal to noise ratio. One aspect that is detrimental to the martensite mIoUs are residual OP-S particle agglomerates at the sample surface. These are surface polishing induced artefacts at the same scale as martensite extrusions and can have a similar shape, as shown by the annotation in Figure 2j. Apart from this, the overall poor model performances on the martensite domain are probably owed to a more pronounced domain gap between ferrite and martensite extrusions. In particular, this is attributed to martensite containing almost exclusively tongue-like extrusions. Furthermore, the damage sites in martensitic steel are much smaller in absolute size relative to the ferritic steel.

**General remarks:** Apart from few exceptions, augmentations improve the average performance across all considered domains relative to the minimal augmented experiments (#8). Most notably elastic distortion (#6) and to some extent optical distortion (#7) boost the overall mIoU in every domain. On the contrary, brightness and contrast augmentation should be treated with care as these seem to potentially cause severe performance drops. Also, it is to be noted that in general, the augmentations impact the target domains more than the source domain.

The elastic distortions represent an essential addition to the augmentation pipeline, irrespective of the material domain. It can possibly be justified by attained (extrusion) shape invariance. For instance, warping the images of ferritic material with its straight slip traces can produce similar image texture observed in copper associated with its wavy slip structures. According to [14], a central role in case of small training sets can be ascribed to elastic transformations. Microscope operators tend to have different preferred brightness and contrast settings. Hence, in terms of image intensity affecting augmentations, it is essential to tune the hyperparameters for an optimal interplay with raw gray value ranges from image acquisition and minimal information loss. Furthermore, the performance decrease (compare #10 and #11 of Table 3) related to downsampling-induced information loss in images is prevalent throughout every material domain and affirms similar observations in literature [26].

The less pronounced mIoU variance in the source domain indicates that extrusions and cracks in its training data set are representative of the ones in its test data set. On the other hand, extrusions from the source domain are not really representative of the ones in the target domains. Hence, the models trained on source domain data set rely more on the careful selection of augmentation types so that it could be generalized well to the extrusions in target domain. On the one hand, this can be achieved if the augmentations on the source domain extrusions render them representative of the ones in target domain. On the other hand, augmentations can force the network not to learn features that are too specific to the source domain, thus learning features that apply to both domains. Whenever augmentations do not help the network to generalize to target domain, the performance drops significantly.

Additional experimentation might be required to find clues on which layer benefits from augmentations specifically. In [27], it has been substantiated that the ability to generalize across domains is layer dependent. While features of initial layers of any two trained networks typically share common characteristics, following layers become increasingly specific and difficult to transfer to alternate domains, unless transfer learning and fine tuning is applied.

The adaptation of the augmentation pipeline holds pronounced potential to improve the foreground class mIoUs of each domain. It can be inferred from the source domain performance that the trained networks, with the right set of augmentations, can cope rather well with loading amplitude and subjective SEM setting induced variance (different working distances, magnifications and brightness/contrast settings). In this work exclusively, SE2 SEM images with a stable perspective were utilized, which posed a simplification to some extent. Therefore, conclusions on the domain generalization to alternate common SEM detector types and different imaging modalities are difficult to draw. Evidently, however, generalizability across material domains can potentially pose a difficult task. A difficulty lies in finding a mutually beneficial augmentation setting for every material domain. Further, even obtaining such an augmentation setting cannot ensure satisfactory domain generalization. In particular, from the model evaluations on different material data sets it can be deduced, that for materials with a more pronounced domain gap to the source domain, i.e., martensite, a different and more elaborate route is required.

#### 3.1.2. Multi-Domain Training

The results from the prior section indicated the demand for additional training in order to be able to generalize over multiple material domains and microstructures with substantial domain gaps. At the same time, results showcased potential domain generalization of models for smaller domain gaps. Hence, training a model with few diverse materials (i.e., damage types) representative for damage types in various materials was pursued over alternative generalization techniques. In our specific case, indications were found suggesting that domain generalization of solely ferrite-trained models to martensite and copper is limited due to ferrite exclusively containing protrusions. Therefore, further training with conjunct data sets adding martensite with tongue-like extrusion morphology and evaluation on all test data sets were performed.

In Table 4, the training settings and corresponding results are described employing the optimized augmentation setting from model #10 in Table 3. Due to the imbalances between ferrite and martensite data sets, see Table 1, few data set imbalance correction (DSIC) strategies were attempted in experiments #3 to #5. Since most scientific problems and thus data sets are either intrinsically or extrinsically imbalanced, there is extensive literature on the imbalance correction, e.g., [28]. In this work initially a sample weight correction (SWC) was tested in which correction factors dependent on the imbalance between training data sets are incorporated into loss calculation. These factors (weights) for ferrite and martensite were computed based on the extrusion pixel ratio between the ferrite and martensite training data set. An alternate route was tested with the sampling correction (SC), where the amount of images sampled from both training data sets was matched by over- and undersampling the martensite and ferrite data set, respectively. These imbalance correction techniques were complemented with a third hybrid approach which combined SWC with over sampling of martensite data set and in the context of this work is referred to as sample weight and sampling correction (SWSC).

**Performance on martensite:** Initially, a target domain (martensite) trained experiment (#1) was evaluated on the martensite test data achieved an mIoU${}_{e}$ of 0.43. Typically target domain trained performances are considered to pose an upper bound [29]. Hence, this reference value exceeds the value observed from the domain generalization of the ferrite trained model (Table 3 experiment #10) by a large margin. While the combined training (#2) is improving mIoU${}_{e}$ on martensite compared to values achieved by domain generalization, it does not quite reach the reference value. This suggests that the dissimilar amount of training images in the ferrite and martensite data sets results in a performance degradation of the jointly trained model on the martensite data set. When imbalance correction is applied, this issue is alleviated. Significant improvements of 0.12, 0.08 and 0.12 on the martensite mIoU${}_{e}$ are observed when SWC (#3), SC (#4) and SWSC (#5) is applied, respectively. When trained on original resolution the SWSC yielded a martensite mIoU${}_{e}$ of 0.47.

**Performance on ferrite:** From the experiments, it can be also be seen that including the martensitic data set for training does not cause significant performance reductions on the ferritic data set. In experiment #2, the slight decline in mIoU${}_{c}$ when compared to Table 3 experiment #10, is compensated by a small improvement in mIoU${}_{e}$. When additionally imbalance corrections in favor of martensite are applied marginal reductions in ferrite performance are noticeable.

With respect to #11 in Table 3, an overall mIoU reduction of 0.02 can be observed for the ferritic data set, while the martensitic mIoU${}_{e}$ boosts by 0.32. In general, it can be inferred that when additionally martensite images were provided to the network during training, a better generalization of extrusions was achieved. After imbalance correction, the network was able to attain segmentation performances for extrusions in ferrite and martensite, which matched the performances of either network dedicated to the respective domains. The imbalance correction represented an essential step to eliminate the networks inclination to learn only extensive protrusions that covered a larger pixel fraction in the data set.

**Performance on copper:** The model trained conjunctly on ferrite and martensite without data imbalance correction (#2) when tested on copper showed an improvement over model #10 from Table 3. It can be observed that this conjunct model is capable of assessing the connectivity of close proximity tongue-like extrusions significantly better than solely ferrite trained models. The issue of merging tongue-like extrusions is alleviated by adding martensite extrusions to the training. However, the balancing methods decreased the copper performance. This could be attributed to strongly weighted tongue-like extrusions shifting the focus away from the protrusions which make up the largest part of the damaged area in copper. To conclude, these observations suggest that there is an optimal balance of ferrite (protrusions) and martensite (tongue-like extrusions) data at which the segmentation performance is maximized for copper.

There is still potential for improvement of model #6 e.g., by optimizing the augmentation pipeline for this multi-domain setting and including the copper data in training. Nonetheless, this model presents a promising starting point, as it achieves satisfactory performance on multiple domains and can be expected to transfer well to a multitude of other metallic materials and imaging conditions. The evaluation of conjunct model’s transferability to further material domains will be subject of future works.

**Future scope:** Alternate approaches for bridging larger domain gaps with sparse target domain data, specifically semi-supervised and unsupervised domain adaptation techniques, are subject to much attention in literature. These techniques rely on annotated source domain data along with few or none annotated images from the target domain, which are passed to specialized architectures. For instance, unsupervised domain adaptation was attempted with a domain-adversarial neural network employing a gradient reversal layer during training in [30]. There are indications that this unsupervised domain adaptation approach can improve the segmentation quality in source and target domain [31].

In [32], the error rate of a domain discriminator network was utilized to quantify domain gap relying only on unlabeled input images of both domains. Further, it was illustrated that the domain gap in conjunction with the error rate of the source task can provide an estimate of how well a network trained in one domain transfers to another domain. This estimate in the future could provide an indication on, whether reusing a trained network on a target material is applicable or there is demand for alternate methods.

#### 3.1.3. Model Evaluation on the Ferrite Domain

In the following, the results of the ferrite trained model performing best on the ferrite test data set (Table 3 #11) are examined in greater detail. The scatter plots in Figure 6, show the tile image-wise mIoU plotted over the corresponding crack/extrusion pixel amount. Complementing this, Figure 7 illustrates a case study of good and poor segmentation instances in Figure 7a–i and Figure 7j–l, respectively.

The characteristic shape of the distribution in Figure 6 can be explained by the fact that extrusions composed of few pixels are usually rather shallow and thus the detection is hampered. In a notable amount of cases faded out boundary sections of extrusions barely extend into a tile which causes poor detectability as well.

While these shallow regions can cause problems for the network, they are typically not critical defects determining fatigue life. Furthermore, the IoU metric penalizes pixel discrepancies in smaller instances more.

The cases in Figure 7a–e show different types of extrusions that are segmented correctly by the network. Considerable changes in texture, brightness and contrast can be observed in these extrusion images. Figure 7a,b show shallow extrusions while cases in Figure 7c–e show larger ones. Figure 7c shows that the presence of multiple slip trace orientations does not impede the segmentation of extrusions. In cases of Figure 7d,e, the network distinguishes the extrusion area from the cracks and crack debris very well.

Similarly, the images of cracks seen in Figure 7f–i show that the network also learned to segment different types of cracks under various imaging conditions. Figure 7f shows a large crack while Figure 7g on the other hand shows a micro crack that has been segmented accurately. In Figure 7h a crack is detected correctly despite regions occluded by crack debris. A crack that is differentiated from the accompanying extrusion can be seen in Figure 7i.

Figure 7j–l represent instances where crack segmentation was not accurate. In Figure 7j previously mentioned fissured protruded areas are depicted. They contain instances of presumably tiny micro cracks, which are on the verge of being considered as intrusions (and hence associated to the extrusion class). Fatigue crack tip friction induced debris covering parts of the crack is shown in Figure 7k. Figure 7l shows a potential inclusion with a darker background concealing cracks and hence impeding its detection.

Generally, pixels for which class affiliation was not obvious were labeled as extrusions. However, as the transition of intrusions to micro cracks is smooth, the discriminability is poor, causing potential labeling inconsistencies. The same applies for occlusions caused by debris where the connectivity assessment of cracks is impeded. While in some instances only visible parts of the crack were labeled as such, in others annotations were based on the assumption that the crack is continuous beneath the debris. There are different techniques reported in the literature to deal with occlusion. For instance, random erasing of pixel regions within the image was suggested in [33] to force the network to take into account the context in the entire image rather than only a small portion. In addition, sample boundaries, which suffer from extreme illumination, are often detrimental to segmentation of extrusions and cracks.

Furthermore, achieved segmentation performances should be improvable by applying other network architectures, advanced data augmentation concepts or supplying more representative data. There are architectures achieving superior IoUs using larger data sets, such as DeepLab v3+ [34], which was optimized in a series of works. It utilizes concepts like atrous convolutions, which upsamples a convolutional filter to increase its field of view while retaining spatial resolution of features, and atrous spatial pyramid pooling which finds features at multiple scales. These architectures might become increasingly relevant for the material scientific community as data sets grow. Research in the field of data augmentation is still ongoing and new techniques including oversampling based augmentation such as neural style transfer [35], generative adversarial networks (GANs) [36] and image mixing [37] flourish. These hold substantial potential to improve domain transfers. Another route for improving segmentation results is test-time augmentations, which can be perceived as an ensemble technique, where multiple predictions of augmented inputs are aggregated. While this method is unfavorable for applications where processing speed is of importance, safety relevant applications, where throughput is secondary such as fatigue, pose an ideal environment for it.

### 3.2. Slip Trace Detection

This section compares and discusses the methods developed for slip trace detection presented in Section 2.3.

A subset of images was handpicked from the ferrite test data set based on the area and uniqueness of the extrusions present in it, constituting a ferrite STO data set. For each extrusion present in this subset, one or multiple expert perceived orientations were labeled. Subsequent to obtaining the segmentation maps from the trained network, all three image processing routines were applied to the extrusion predicted regions. A parameter optimization was conducted for each method on the basis of this data set. Then, the computed extrusion STO $\phi_{pred}$ from each method is compared against the expert perceived orientation labels $\phi_{perc}$ for each image, as represented for an image in Figure 8a.

The graph in Figure 8b shows the correlation of the orientation detected from the three methods with the perceived orientations for all images in the ferrite STO data set. It becomes evident, that the three methods match the perceived orientation well. Additionally, it can be observed from the standard deviation of the error ($\sigma_{x}$) that the ICEO method achieved the best performance, followed by CHL, while CSO scored the worst result. The orientations observed in Figure 8b indicate that most extrusions STOs in this subset are concentrated in the bands 60–80${}^{\circ }$ and 100–120${}^{\circ }$.

The authors believe that exploiting extrusion masks and probabilities predicted from the segmentation network in all three methods significantly improved the accuracy of the predicted STOs. This is achieved by discarding regions of intensity variations arising from cracks, pores, scratches, particles and grain boundaries in the image. The extrusions present in the STO data set can be classified in two groups. There were extrusions like in Figure 4b, where the STOs were spatially spread in a disordered manner, with a moderate spread in orientation (*disordered extrusions*). On the other hand, there were *ordered extrusions* like in Figure 9a, where the predominant STOs were distinct spatially and in orientation. In ordered extrusions these major STOs may differ only mildly in orientation. The superior performance of the ICEO method can be ascribed to its versatility. It entailed separate clustering steps for orientation and centroids. A stricter threshold for the orientation clustering enables that in ordered extrusions, the predominant STOs that differ only mildly in orientation could be distinguished, like the clusters that are shown in Figure 9b. For a disordered extrusion, a strict orientation clustering results in detection of multiple orientation clusters, as shown in Figure 5g. However, the subsequent iterative spatial clustering step ensures merging of clusters if such orientation clusters are not spatially well separated and differ only moderately in orientation, as shown in Figure 5h. Thus, ICEO tackles both ordered and disordered extrusions and hence is expected to work well for different materials, irrespective of the slip characteristics.

The CHL method had the disadvantage that spatial clustering cannot be accurately performed on the extruded regions since they are represented as Hough lines. This led to the restriction that a single orientation threshold had to be optimized for all extrusions in the ferrite STO data set. Hence, unlike ICEO, stricter orientation thresholds could not be set. This resulted in cases as shown in Figure 9c, where the cluster coloured red includes the two major STOs in that ordered extrusion. Moreover, the Hough line detection gives more preference to longer edges and do not enforce edge connectivity. Hence, individual longer edges in the extrusion can contribute to multiple Hough lines and thus be wrongly detected as a STO, as shown by the yellow clusters in the aforementioned figure.

The CSO method’s main disadvantages were in the extraction of extruded regions. Its strong reliance on the probability mask from the network did not always fetch good results, since the network need not necessarily give a higher probability value to the extruded regions in its predictions. Moreover, the method’s dependence on a single threshold on the gradient magnitude was less efficient in detecting the extruded regions. Liberal gradient thresholds meant that false regions are detected, while stricter thresholds resulted in detecting only parts of the extruded regions. On the other hand, the canny algorithm, used by the other two approaches, uses an iterative thresholding, which ensured that one can set a stricter threshold to select high intensity gradient regions, while also including neighboring regions that are above a second threshold.

Another interesting inference from Figure 8b is the distribution of orientations in the STO data set. The authors believe that this bimodal distribution is representative of the whole ferrite data set. This STO distribution is affected by the out-of-plane bending loading as well as the texture and slip properties of the polycristalline material.

## 4. Summary and Conclusions

Fatigue damage accumulation processes are determined by many interdependent mechanisms on various size- and time-scales. These dependencies need to be addressed by computational and data-driven fatigue modeling efforts. As statistical validation of crack initiation events predicted from micromechanical fatigue simulations is demanded, accurate, reliable and automated methods are required to localize experimentally observed damage locations.

Therefore, the initial objective was to verify the feasibility of DL-based early fatigue damage segmentation. As a prerequisite to achieve the aforementioned objective, three distinct data sets based on ferritic steel, copper and martensitic steel SE2 SEM images were created. Furthermore, in order to handle lack of training data a custom augmentation pipeline was designed. Semantic segmentation of extrusions and micro cracks with an U-Net is demonstrated. The segmentation on the ferrite data set achieved mIoU values of 0.84 and 0.71 for the foreground classes crack and extrusion, respectively, proving the feasibility of DL-based damage segmentation.

A decisive criterion on whether the method can be established to validate damage formation and crack initiation/growth models is whether it can generalize to other imaging conditions and material systems. The transferability and generalization of these DL models reduces labeling and training efforts and thus represents a prerequisite for applying them to a wide range of materials in an efficient manner. This challenge was addressed by testing whether an augmented ferritic steel trained model can generalize to copper and martensitic steel. These materials exhibited characteristic and distinct extrusion morphology. According to the computed mIoUs, the ferrite model could generalize to copper while extrusions in martensite presented a challenge. Additional visual inspection suggested that this can be traced back to the domain distance between source and target material domain. The copper extrusions are to a large fraction extensive extrusions (protrusions) which are comparable to the ones observed in ferrite. A minor fraction of extrusion area in copper is represented by highly localized tongue-like extrusions. In contrast martensite extrusions were almost exclusively tongue-like. These tongue-like extrusions in both target material domains were poorly predicted by the ferrite trained model as these were barely present in the ferritic steel data set.

In order to ensure generalization of a model to a wide range of materials two more elaborate concepts are imaginable. On the one hand, there are promising developments in the field of domain adaptation techniques to cope with material related differences. On the other hand, the supervised training of a model from scratch with few characteristic material classes seems feasible. Indeed, combined supervised training with ferrite and martensite data sets showed a substantial improvement over sole augmented ferrite training on the martensite test set, especially when sample imbalance correction methods were employed. At the same time the good performance on the ferrite test data set was maintained. This indicates that networks can attain a generalized grasp of extrusions. Further, including martensite damage locations in the training, assisted the copper damage segmentation as additional segmentation of the fewer tongue-like extrusions in copper was enabled. In summary, when aiming for prediction of extrusions with a set of distinct morphology, training images are required to contain corresponding extrusion types. If this requirement was not satisfied, the applied data augmentations, while impacting the results substantially, could not facilitate satisfactory material domain generalization. In contrast, imaging condition induced variance arising from operator subjectivity could be handled by the data augmentations.

Since the plastic anisotropy and active slip systems need to be modeled accurately, automatic determination of slip trace orientations posed another objective of this work. In this regard, an approach relying on iterative clustering of Canny edge orientations showed the best agreement with expert annotated orientations among the three presented approaches. The methods also made use of the extrusion probability mask retrieved from the segmentation network.

From the authors’ point of view, automated DL-based segmentation of extrusions and cracks is an essential step to statistically validate micromechanical damage initiation models. These validations are indispensable in order to predict fatigue damage localization and thus fatigue life accurately. Therefore, these tools hold the potential to assist, improve and accelerate product and material design with superior mechanical characteristics.

Apart from the fatigue domain, these segmentation methods can potentially be applied to a wide range of material scientific problems including tribological friction marks, fracture surface analysis, metallographic constituent determination, corrosion and polymer crazing. Thus, these data-driven methods represent a promising versatile tool to obtain statistical material representations required for the digitization of materials.

## Figures and Tables

**Figure 1 materials-13-03298-f001:**
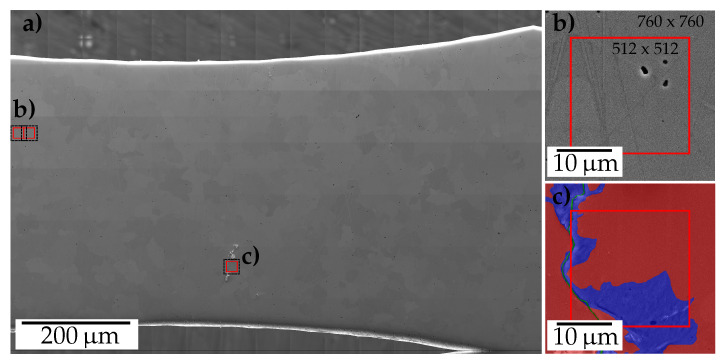
Overview stitched high-resolution secondary electron SEM image of a ferritic sample surface in (**a**). (**b**) Sample tile at the boundary indicating the original tile size and reduced labeling mask size retrieved from the network. The mirror padding applied at the boundary becomes apparent as well. (**c**) A tile at a short crack surrounded by extruded volume overlaid with the corresponding labeling mask displayed as a one-hot encoded RGB color image.

**Figure 2 materials-13-03298-f002:**
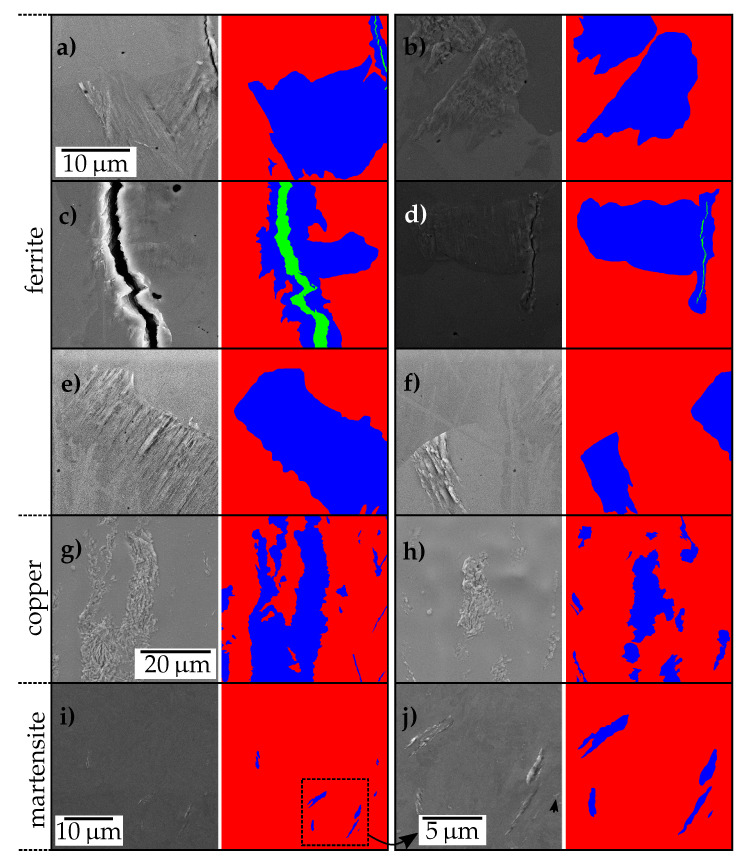
Glimpse at the ferrite (**a**–**f**), copper (**g**,**h**) and martensite (**i**,**j**) extrusion and crack data sets and their corresponding labeling masks. All images from the same material are at the same scale, with the exception of (**j**), which is a magnified view of (**i**). The set of images shows the variable brightness, contrast conditions and extrusion and crack geometries.

**Figure 3 materials-13-03298-f003:**
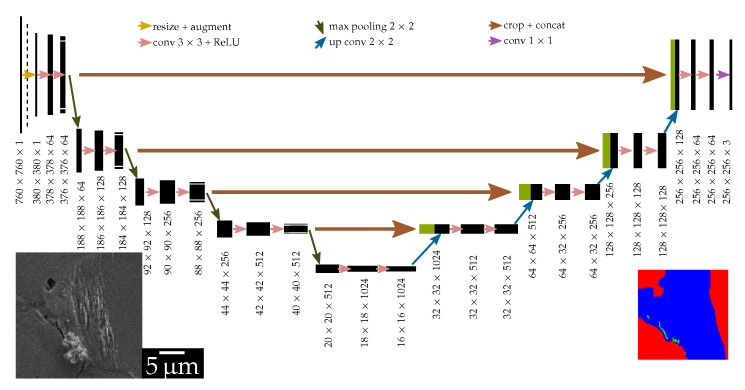
U-Net architecture implemented consisting of four levels with 3 × 3 convolutions and max pooling layers on the encoder section, and up-convolution and 3 × 3 padded convolutions on the decoder section (after [14]).

**Figure 4 materials-13-03298-f004:**
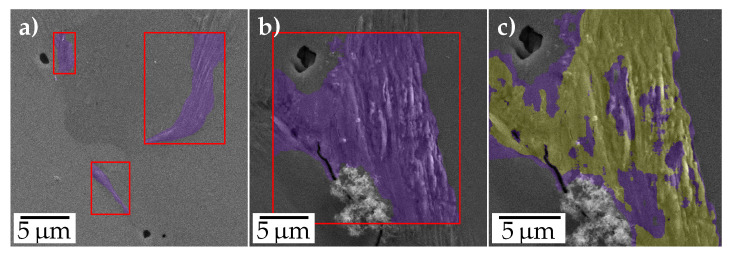
Extrusion images extracted using the extrusion masks from semantic segmentation for two example images (**a**,**b**), and the extrusion image from (**b**) overlaid with a thresholded extrusion probability mask (**c**).

**Figure 5 materials-13-03298-f005:**
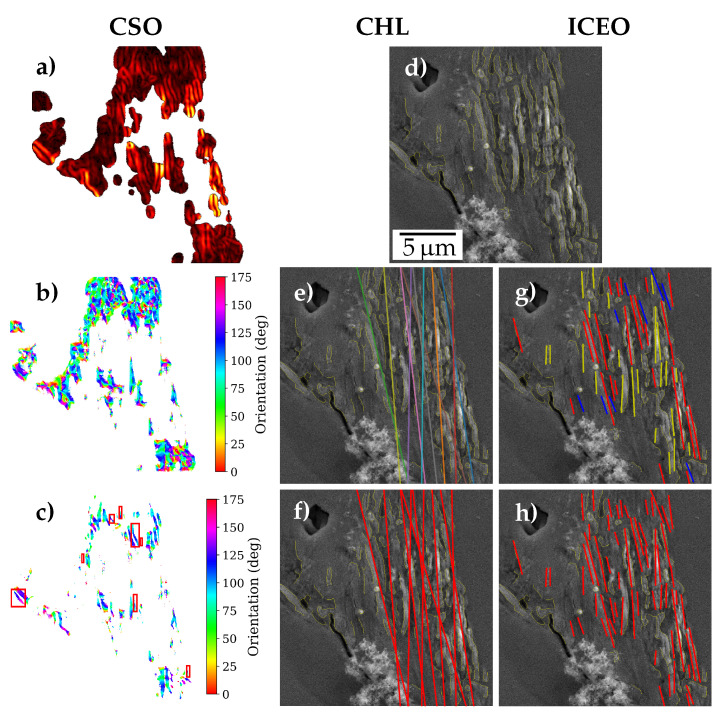
Overview of the methods to detect slip trace orientations (STOs) (illustrated based on Figure 4b): For the clustering of Scharr gradient based edge orientations (CSO), gradient (**a**) magnitude and (**b**) orientation of high confidence region, and (**c**) the selected orientation blobs after thresholding and filtering are depicted. Image (**d**) shows filtered skeletons after Canny and morphological operations, which are common for clustering Hough lines from edges (CHL) and iterative clustering of edge orientations (ICEO). (**e**) The Hough lines detected and (**f**) clustered in CHL are shown. (**g**) Clusters after orientation-based clustering in ICEO and (**h**) after centroid-based iterative merging of clusters are illustrated. All figures have the same scale.

**Figure 6 materials-13-03298-f006:**
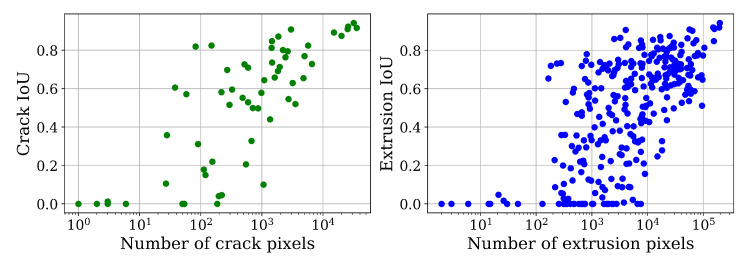
Tile image-wise IoU metric shown as a function of contained crack and extrusion pixels, respectively.

**Figure 7 materials-13-03298-f007:**
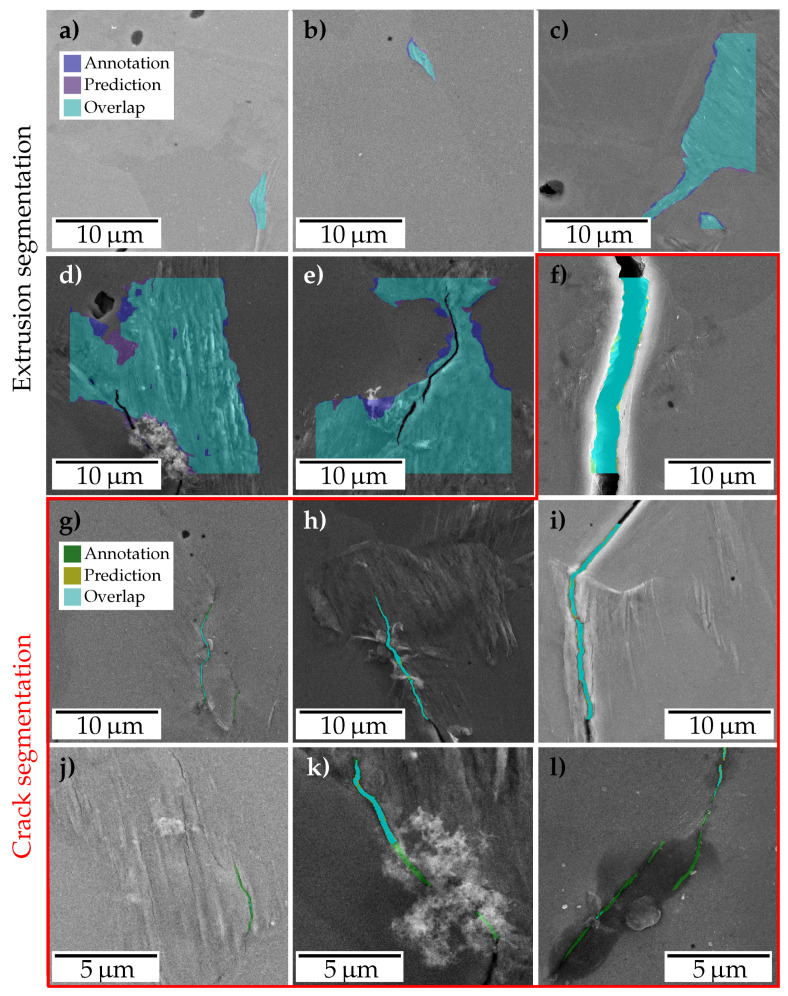
Examples for common cases of extrusion and crack segmentations. Note that the border regions of images (**a**–**i**) neither show annotations nor predictions since the latter is not provided by the network in these regions. Images (**j**–**l**) are an exception to this as they show a subset of pixels.

**Figure 8 materials-13-03298-f008:**
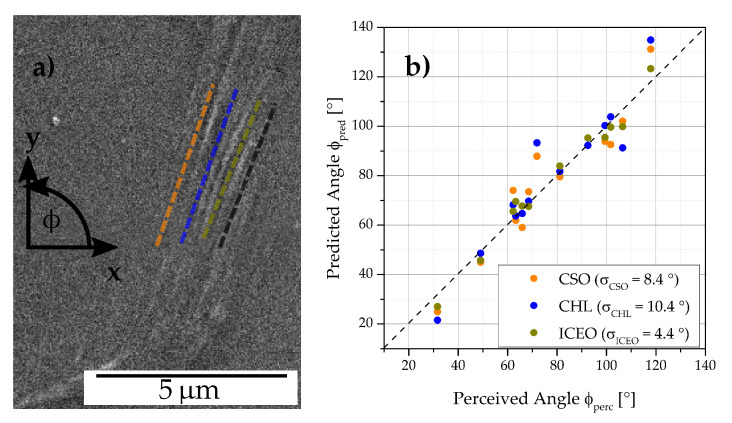
In (**a**) the expert perceived orientation $\phi_{perc}$ (black) as well as different predicted slip trace orientations $\phi_{pred}$ derived using clustering of Scharr gradient based edge orientations (orange), clustering Hough lines from edges (blue) and iterative clustering of edge orientations (green) are illustrated for an example extrusion. The correlation of these $\phi_{pred}$ with $\phi_{perc}$ for a set of extrusions is shown in (**b**). Further, the standard deviation of the error ($\sigma_{x}$) for each method is given.

**Figure 9 materials-13-03298-f009:**
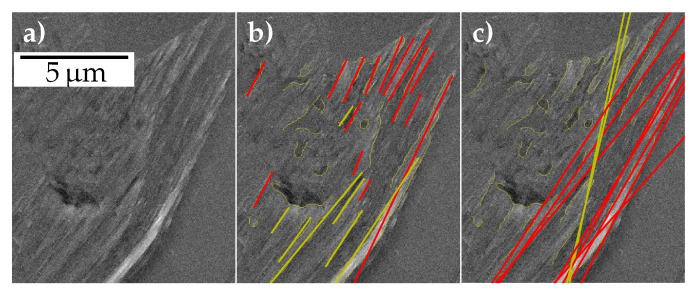
Images comparing STO results from ICEO and CHL. (**a**) extrusion image and (**b**,**c**) slip trace orientation clusters found from using ICEO and CHL, respectively.

**Table 1 materials-13-03298-t001:** Overview of data sets.

	Ferrite		Copper	Martensite	
Metric	Training	Testing	Testing	Training	Testing
N [-]	12		1	3	
N${}_{t}$ [-]	3940	860	357	763	168
p${}_{c}^{i}$ [%]	5.35	6.97	-	-	-
p${}_{e}^{i}$ [%]	33.70	34.30	89.92	36.56	33.92
p${}_{c}^{p}$ [%]	0.06	0.09	-	-	-
p${}_{e}^{p}$ [%]	2.34	2.21	12.13	0.18	0.17
P${}_{c}^{p80}$ [%]	1.15	2.00	-	-	-
P${}_{e}^{p80}$ [%]	10.75	10.00	20.45	0.62	0.63

**Table 2 materials-13-03298-t002:** Description of individual augmentations implemented using the albumentations framework.

Augmentation Type	Description: Aug. Type/Parameter(s)	x${}_{lim}$	*p*
Affine transformation	Linear transformation/rotate, shift and scale limit	30, 0.1, 0.1	0.8
Rotation 90${}^{\circ }$	-/-	-	0.25
Reflection	-/-	-	0.25
Elastic transformation	Local deformations/alpha affine, alpha, sigma, approx.	0, 40, 6, True	0.4
Optical distortion	Barrel or pincushion/distort limit, shift limit	0.1, 0.5	0.25
Gaussian blurring	Convolution Gaussian kernel/blur kernel size	7	0.2
Motion blurring	Convolution motion-blur kernel/blur kernel size	3	0.2
Gaussian noise	-/var limit	0.015	0.4
Contrast	-/limit	0.15	0.4
Brightness	-/limit	0.1	0.4

**Table 3 materials-13-03298-t003:** Results of augmentation pipeline optimization. Note that even in experiment #8, non-mutual augmentations between this table and Table 2 are applied. (s) and (t) denotes source and target domains, respectively.

Experiment #	Resolution	Brightness	Contrast	Gaussian Blurring	Motion Blurring	Gaussian Noise	Elastic Transformation	Optical Distortion	Ferrite mIoU${}_{c}$ (s)	Ferrite mIoU${}_{e}$ (s)	Ferrite mIoU${}_{o}$ (s)	Copper mIoU${}_{e}$ (t)	Martensite mIoU${}_{e}$ (t)
1	N · 0.25	**✓**	**✗**	**✗**	**✗**	**✗**	**✗**	**✗**	0.75	0.49	0.74	0.55	0.05
2		**✗**	**✓**	**✗**	**✗**	**✗**	**✗**	**✗**	0.77	0.55	0.77	0.37	0.06
3		**✗**	**✗**	**✓**	**✗**	**✗**	**✗**	**✗**	0.74	0.52	0.75	**0.68**	0.07
4		**✗**	**✗**	**✗**	**✓**	**✗**	**✗**	**✗**	0.76	0.51	0.75	0.66	0.05
5		**✗**	**✗**	**✗**	**✗**	**✓**	**✗**	**✗**	0.70	0.51	0.73	0.67	0.13
6		**✗**	**✗**	**✗**	**✗**	**✗**	**✓**	**✗**	0.76	0.52	0.76	0.67	**0.17**
7		**✗**	**✗**	**✗**	**✗**	**✗**	**✗**	**✓**	0.76	0.50	0.75	0.67	0.09
8		**✗**	**✗**	**✗**	**✗**	**✗**	**✗**	**✗**	0.75	0.52	0.75	0.51	0.06
9		**✓**	**✓**	**✓**	**✓**	**✓**	**✓**	**✓**	**0.82**	0.60	0.80	0.33	0.14
10		**✗**	**✓**	**✓**	**✓**	**✗**	**✓**	**✓**	0.81	**0.66**	**0.82**	0.26	0.14
11	N	**✗**	**✓**	**✓**	**✓**	**✗**	**✓**	**✓**	0.84	0.71	0.85	0.61	0.15

**Table 4 materials-13-03298-t004:** Results of different training and testing data sets including data set imbalance correction schemes. The letters m and f in the training set denote martensite and ferrite, respectively.

Experiment #	Resolution	Training Set	DSIC Type	Ferrite mIoU${}_{c}$	Ferrite mIoU${}_{e}$	Ferrite mIoU${}_{o}$	Martensite mIoU${}_{e}$	Copper mIoU${}_{e}$
1	N · 0.25	m	-	-	-	-	**0.43**	-
2		m + f	-	**0.82**	**0.68**	**0.83**	0.31	**0.38**
3		m + f	SWC	**0.82**	0.62	0.81	**0.43**	0.32
4		m + f	SC	0.80	0.64	0.81	0.39	0.27
5		m + f	SWSC	0.79	0.62	0.80	**0.43**	0.31
6	N	m + f	SWSC	0.83	0.67	0.83	0.47	0.58

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
