# Peer review of "Automated Quantitative Analyses of Fatigue-Induced Surface Damage by Deep Learning"

_materials, 2020, doi:10.3390/ma13153298_

Round 1

Reviewer 1 Report

This article talks about the possibility of implementing deep learning (DL) methodologies for semantic segmentation and different image processing approaches for quantitative slip trace characterization and generalizes this methodology to multiply materials, which shows promising results. 

1. Why selecting ferrite steel, copper, and martensite steel as testing samples over other materials? Can you further elaborate on your selection? 

Author Response

Dear reviewer,

thank you for your suggestions. Besides English improvements over the whole document, we addressed the following questions.

  1. Why selecting ferrite steel, copper, and martensite steel as testing samples over other materials? Can you further elaborate on your selection?

We included/adjusted some parts in the data set section (line 126-129), see attached document. There the following color code is used.

blue = inserted

orange = replaced

red = removed

Reviewer 2 Report

Dear Authors,

1) In Summary and conclusion, I understand that the method established for ferrite extrusion can generalize well for copper but cannot generalize so well for martensite. Are the conclusions based only on the mIoU numbers in Table 3 and 4?
2) In Figure 8, the angles are shown. What is the reference direction for the angles? What do the σ’s stand for?
3) In Figure 7, blue, cyan, green regions exist but no violet or yellow regions exist. The legends can help confused readers.
4) In Figure 6, the image-wise IoUs for the ferrite-trained model are presented. Why are no image-wise IoUs for copper and martensite presented?
5) In Figure 4, what is the relationship between a) and b)? Figure 4b-c, Figure 5 and Figure 7d are from the same region?

Author Response

Dear reviewer,

thank you for your suggestions. Besides English improvements suggested by reviewer 1 over the whole document, we addressed the following questions.

1) In Summary and conclusion, I understand that the method established for ferrite extrusion can generalize well for copper but cannot generalize so well for martensite. Are the conclusions based only on the mIoU numbers in Table 3 and 4?

We have added some parts in line 675-676 (see attached document) which address this question of yours.

2) In Figure 8, the angles are shown. What is the reference direction for the angles? What do the σ’s stand for?

We have added some parts in the figure 8 caption and adjusted the images in figure 8 slightly to show a reference coordinate system (see attached document).

3) In Figure 7, blue, cyan, green regions exist but no violet or yellow regions exist. The legends can help confused readers.

That's a good idea. We have added a legend in figure 7 for the color schemes in extrusion segmentation and crack segmentation, respectively (see attached document).

4) In Figure 6, the image-wise IoUs for the ferrite-trained model are presented. Why are no image-wise IoUs for copper and martensite presented?

Because if we would have included these scatter plots for the other materials further elaborate analysis and example segmentation would be required for completeness. This would render the document substantially longer. We rather intend to write a detailed follow-up publication where we focus even more on domain generalization. 

5) In Figure 4, what is the relationship between a) and b)? Figure 4b-c, Figure 5 and Figure 7d are from the same region?

That's correct, they are from the same region, we added a note where we believed it to be relevant (figure 4 and 5 are now linked). These two figures can be seen as examples to illustrate the methods rather than results. Linking them to figure 7, which is a result, is not necessary.

In the attached document the following color code is used.

blue = inserted

orange = replaced

red = removed

Reviewer 3 Report

The article has employed deep learning for semantic segmentation to identify cracks and surface extrusions. The topic is original and highly interesting to the fatigue damage modeling research and development community. Other than that the abstract and conclusion are a bit wordy and long, the whole manuscript is written in good English language and style and the results are clearly presented. Publication is recommended.

Author Response

Dear reviewer,

thank you for your kind response. Due to the pronounced interdisciplinarity, we tried to make the abstract and summary understandable for material and computer scientists. In the process it became lengthy, but we think it is required also to provide a self-contained description of the work. We have attached a pdf document summarizing the changes suggested by the other reviewers to this cover letter.

In the attached document the following color code is used.

blue = inserted

orange = replaced

red = removed

Best regards

Ali/Akhil

Reviewer 4 Report

The submitted manuscript entitled ‘Automated quantitative analyses of fatigue induced surface damage by deep learning’ deals with the analysis of surface defects in the aspect of fatique by dee learning methods applied on numerous photographs identifying the defects.

The manuscript is really interesting; however, it is very difficult to judge since it is also very multidisciplinary on the border between materials testing and computational methods.

Because of the high level and deepness of the manuscript, it is recommended for publication at least in the oinion of this Reviewer, but a short minor issues also arose.

- Add scalebars to subgifs 1a and 1b. Please also label the main image starting by a).

- Please add scalebars to all images in fig 2., please refer to the labels in the caption.

- Please add scalebars to figs 4., 5., 7., 8a and 9.

Author Response

Dear reviewer,

thank you for the kind words and suggestions.

1) Add scalebars to subgifs 1a and 1b. Please also label the main image starting by a).

We did it as you suggested, please see attached document.

2) Please add scalebars to all images in fig 2., please refer to the labels in the caption.

We did it as you suggested, please see attached document. In figure 4 we used only one scalebar per material as they are identical for all images of each material. We included a note in the caption.

3) Please add scalebars to figs 4., 5., 7., 8a and 9.

We did it as you suggested, please see attached document.

In figure 5 we used only one scalebar as it is identical for all images. We included a note in the caption. For figure 7 we additionally added legends as suggested by another reviewer.

In the attached document the following color code is used.

blue = inserted

orange = replaced

red = removed
